# On the Use of Hemadsorption with CytoSorb in Patients with Septic Shock. Comment on Kogelmann et al. First Evaluation of a New Dynamic Scoring System Intended to Support Prescription of Adjuvant CytoSorb Hemoadsorption Therapy in Patients with Septic Shock. *J. Clin. Med.* 2021, *10*, 2939

**DOI:** 10.3390/jcm11020334

**Published:** 2022-01-11

**Authors:** Alexander Supady, Philipp M. Lepper, Daniel Duerschmied, Tobias Wengenmayer

**Affiliations:** 1Department of Medicine III (Interdisciplinary Medical Intensive Care), Medical Center, Faculty of Medicine, University of Freiburg, Hugstetter Strasse 55, 79106 Freiburg, Germany; daniel.duerschmied@uniklinik-freiburg.de (D.D.); tobias.wengenmayer@uniklinik-freiburg.de (T.W.); 2Department of Cardiology and Angiology I, Heart Center, University of Freiburg, Hugstetter Strasse 55, 79106 Freiburg, Germany; 3Heidelberg Institute of Global Health, University of Heidelberg, 69117 Heidelberg, Germany; 4Department of Internal Medicine V—Pneumology, Allergology and Critical Care Medicine, Saarland University Medical Center and University of Saarland, 66421 Homburg, Germany; philipp.lepper@uks.eu

With great interest we read the article by Klaus Kogelmann and co-authors on the “First Evaluation of a New Dynamic Scoring System Intended to Support Prescription of Adjuvant CytoSorb Hemoadsorption Therapy in Patients with Septic Shock” [1]. The authors report interesting data from a retrospective evaluation of 502 patients in septic shock treated in four centers in Germany; 198 of these patients received adjunctive CytoSorb treatment. The sheer number of patients is impressive—to our knowledge, this is the largest reported cohort of septic patients treated with CytoSorb, so far.

The authors developed a “Dynamic Scoring System” to assess the prognosis of patients in septic shock and for assistance in selecting patients suitable for adjunctive cytokine adsorption therapy. However, it remains unclear how the score was designed, and how the different parameters were selected. The performance of the score was not tested in the derivation cohort, and no numbers illustrating the performance of the score were given. When reporting and utilizing a clinical score, it should be validated internally or externally [2,3]. In the context of sepsis, a score should be compared to existing scoring systems, most importantly the sequential organ failure assessment (SOFA) score to understand the performance and added benefit of this new score [4]. This comparison is missing, the SOFA score was not reported in this work. To our understanding, rather than developing a score, the authors divided patients into three distinct groups according to a variety of parameters, all describing clinical instability.

With respect to CytoSorb treatment, important treatment information remains unclear and must be reported. The authors did not explain selection criteria for patients treated with CytoSorb. Furthermore, the duration of cytokine adsorption and average duration of use of the adsorbers should be reported.

The primary assessment in this study is based on the comparison of patients treated with or without CytoSorb. Therefore, in addition to Table 1, baseline characteristics for these two groups should also be reported allowing to detect and understand potential differences between these two groups at baseline. Furthermore, it would be interesting to learn how many patients were recruited from the different participating centers and during what time-period the patients were treated. The authors should also describe standard of care for sepsis in these centers and differences between the centers. The comparison group without CytoSorb is ill-defined and the authors should explain why matching has not been performed.

The authors report results from a multivariate logistic regression model and conclude, “that the use of the CytoSorb device reduced the odds of mortality at day 56 by 44.8%”. This interpretation of the results is not sufficiently supported by the reported data—the 95% confidence interval for the reported odds ratio of survival on day 56 for the treatment with CytoSorb of 0.552 was 0.275–1.108. These findings do not translate into a significant treatment benefit of CytoSorb with respect to survival until day 56 in this cohort.

Finally, in the limitations section the authors describe that “there were a number of patients with a ‘do not resuscitate’ (DNR) order in the data sets, which were consequently not included”. The authors should report more details on these patients, especially the number of these patients, and give information on CytoSorb treatment in this group, since this may have relevant implications for the interpretation of the reported results.

In summary, the added benefit of the “Dynamic Scoring System” introduced by the authors remains unclear due to a lack of information on the choice of parameters and due to a lack of validation. Furthermore, the reported results of this retrospective analysis do not allow for a meaningful statement for whether adjuvant therapy with CytoSorb was beneficial for these patients with respect to survival or not. The reported findings from this analysis should not be interpreted in favor of early use of CytoSorb in patients with septic shock.

A previous randomized study on the use of CytoSorb in septic patients could not find a benefit of this therapy [5]. Therefore, additional data from larger randomized trials should be awaited before the widespread use of CytoSorb in the treatment of sepsis outside of controlled trials can be recommended. Until then, we believe a comparison of the here described large cohort of patients treated with CytoSorb with propensity matched controls would be very interesting.

## Data Availability

Not applicable.

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
