# Peer review of "On the Use of Hemadsorption with CytoSorb in Patients with Septic Shock. Comment on Kogelmann et al. First Evaluation of a New Dynamic Scoring System Intended to Support Prescription of Adjuvant CytoSorb Hemoadsorption Therapy in Patients with Septic Shock. J. Clin. Med. 2021, 10, 2939"

_jcm, 2022, doi:10.3390/jcm11020334_

Round 1
Reviewer 1 Report
In this commentary, Supady and colleagues made a statement on the use of hemadsorption with CytoSorb in septic shock patients which was shown in the article recently published in J Clin Med (Kogelmann et al.). The authors acknowledged the number of septic- shock patients subjected to adjunctive CytoSorb treatment but pointed out insufficient information for the controls, parameters and validation used in “Dynamic Scoring System”. Also, the authors indicated that the data obtained from this retrospective analysis may not possess any authenticity for potentiating the clinical benefit of CytoSorb treatment. The authors included a recommendation for no ignorance of reporting information on CytoSorb treatment to DNR patients and encouraged additional data from larger randomized trials. To my understanding, this commentary has been well written and sounds scientifically reasonable. I do not have any additional review comments on this commentary.